

# Detecting heterogeneity in single-cell RNA-Seq data by non-negative matrix factorization

Xun Zhu[1,2], Travers Ching[1,2], Xinghua Pan[3], Sherman M. Weissman[3] and Lana Garmire[1]

[1] Epidemiology Program, University of Hawaii Cancer Center, Honolulu, HI, United States
[2] Molecular Biosciences and Bioengineering Graduate Program, University of Hawaii at Manoa, Honolulu, United States
[3] Department of Genetics, Yale University, New Haven, CT, United States

## ABSTRACT

Single-cell RNA-Sequencing (scRNA-Seq) is a fast-evolving technology that enables the understanding of biological processes at an unprecedentedly high resolution. However, well-suited bioinformatics tools to analyze the data generated from this new technology are still lacking. Here we investigate the performance of non-negative matrix factorization (NMF) method to analyze a wide variety of scRNA-Seq datasets, ranging from mouse hematopoietic stem cells to human glioblastoma data. In comparison to other unsupervised clustering methods including K-means and hierarchical clustering, NMF has higher accuracy in separating similar groups in various datasets. We ranked genes by their importance scores ($D$-scores) in separating these groups, and discovered that NMF uniquely identifies genes expressed at intermediate levels as top-ranked genes. Finally, we show that in conjugation with the modularity detection method FEM, NMF reveals meaningful protein-protein interaction modules. In summary, we propose that NMF is a desirable method to analyze heterogeneous single-cell RNA-Seq data. The NMF based subpopulation detection package is available at: https://github.com/lanagarmire/NMFEM.

Corresponding author
Lana Garmire,
LGarmire@cc.hawaii.edu

## INTRODUCTION

The advancement of technologies has enabled researchers to separate individual cells from bulk and sequence their transcriptomes at the single cell level, known as single-cell RNA-Sequencing (scRNA-Seq). This technology has reached an unprecedented fine resolution to reveal the program of gene expression within cells (*Kumar et al., 2014*). It was used to detect heterogeneity within the cell population, and it has greatly enhanced our understanding of the regulatory programs involved in systems such as glioblastoma (*Patel et al., 2014*), neuronal cells (*Usoskin et al., 2014*), or pluripotent stem cells (PSCs) (*Kumar et al., 2014*). It was also used to delineate cell types and subpopulations in differentiating embryonic cells (*Treutlein et al., 2014*). Other applications include uncovering multilineage priming processes involved in the initial organogenesis (*Brunskill et al., 2014*), and substantiating

the hypothesis of inter-blastomere differences in 2- and 4-cell mouse embryos (*Biase, Cao & Zhong, 2014*). Indeed, scRNA-Seq has already made profound impacts on our understanding of the diversity, complexity, and irregularity of biological activities in cells. It will continue to provide more transformative insights in the near future (*Pan, 2014*; *Poirion et al., 2016*).

However, relative to the experimental technology, the bioinformatics tools to analyze scRNA-Seq data are still lagging behind. Recently, various methods have been developed to detect subpopulations (or sub-clusters) within a group of cells using scRNA-Seq data, including scLVM (*Buettner et al., 2015*), BackSpin (*Zeisel et al., 2015*), PAGODA (*Fan et al., 2016*), and SEURAT (*Macosko et al., 2015*). These new computational tools are evidence that understanding scRNA-Seq heterogeneity is of paramount importance. Moreover, once the subpopulations are identified, it is very crucial to find the gene expression signatures that are characteristic of each subpopulation (subclass), in order to reveal the subline biological mechanisms.

Population-level RNA-Seq differential expression analysis, such as DESeq2 (*Love, Huber & Anders, 2014*) and edgeR (*Robinson, McCarthy & Smyth, 2010*), are designed to compare pre-labeled classes. However, it is questionable if they are desirable to identify subpopulations in scRNA-Seq data. Recently, a couple of methods have been reported in the scRNA-Seq analysis domain (*Brennecke et al., 2013*; *McDavid et al., 2013*; *Kharchenko, Silberstein & Scadden, 2014*). For example, a statistical variance model based on gamma distribution was developed to account for the high technical noise occurring in scRNA-seq experiments, such that genes with high squared correlation of variations ($CV^2$) relative to mean expression were identified as "significantly differentially expressed" between two conditions (*Brennecke et al., 2013*). Another Bayesian approach was proposed for scRNA-Seq differential expression analysis, by utilizing a probabilistic model of expression-magnitude distortions that were commonly observed in noisy single-cell experiments (*Kharchenko, Silberstein & Scadden, 2014*). This method later was used for classification of sensory neurons using scRNA-Seq (*Usoskin et al., 2014*). On the other hand, an R package, Monocle, was developed recently for single-cell lineage construction (*Trapnell et al., 2014*).

Previously, NMF has been applied to other areas in computational biology, such as molecular pattern discovery (*Brunet et al., 2004*), class comparison and prediction (*Gao & Church, 2005*), cross-platform and cross-species analysis (*Tamayo et al., 2007*), and identification of subpopulations of cancer patients with mutations in similar network regions (ref). More recently, NMF has been applied to gene expression profiling studies at the population level (*Qi et al., 2009*). Compared to other methods, it showed multiple advantages, such as less sensitivity to *a priori* selection of genes or initial conditions and the ability to detect context-dependent patterns of gene expression (*Rajapakse, Tan & Rajapakse, 2004*). Based on these properties, we hypothesize that NMF is less prone to the influence of noise in the scRNA-Seq data, and thus it can detect a group of genes that robustly differentiate single cells from different subpopulations. In this report, we demonstrate the capabilities of NMF in scRNA-Seq data analysis in these following aspects: (1) accurate clustering of single cells from different conditions in an unsupervised manner; (2) detection of important genes associated with differences among subclasses

(subpopulations); (3) identification of protein-protein interaction modules around the important genes, based on a modified implementation of Functional Epigenetic Modules (FEM) (*Jiao, Widschwendter & Teschendorff, 2014*). We organized the NMF workflow into a streamlined modularity detection R package called NMFEM.

## MATERIALS & METHODS

### Datasets

Five scRNA-Seq datasets are used in this study. For the first four datasets that don't have UMI, normalized fragments per kilo-base per million reads (FPKM) is calculated and used as input to the clustering methods. For the last dataset (iPSC) where UMI technique is adopted, we used the author provided "molecule counts" as the input.

### Mouse lung epithelial cells

scRNA-Seq data were retrieved from the original 201 samples of lung distal epithelial cells of embryonic mouse (GSE52583) (*Treutlein et al., 2014*). The original dataset contains cells collected from four time points: E14.5, E16.5, E18.5 and AT2. After filtering (the procedure is detailed in the "Sample and Gene Filtering" section below), 201 single-cell samples and 4,594 genes are used in downstream analysis. The closest pair of groups, which consists of 45 cells from E14.5 and 27 cells from E16.5 was used for unsupervised clustering comparison.

### Mouse HSCs and MPP1s

scRNA-Seq data were extracted from mouse hematopoietic stem cells (HSCs) and early multipotent progenitors (MPP1s). The data were pre-processed into the format of a FPKM expression profile (Table S1), which include 59 HSCs and 53 MPP1 single cells. After filtering, 95 samples and 2,887 genes were used in downstream analysis.

### Glioblastoma

scRNA-Seq data were retrieved from the original 875 samples of glioblastoma tumor cells in 5 patients (MGH26, MGH28, MGH29, MGH30, and MGH31), along with population and cell line controls (GSE57872) (*Patel et al., 2014*). After filtering, 419 single-cell samples and 6,996 genes were used in downstream analysis. The closest pair of groups, which consists of 80 cells from patient MGH29 and 73 cells from MGH31 was used for unsupervised clustering comparison.

### Mouse bone marrow

scRNA-Seq data were extracted from mouse macrophage DC progenitors (MDPs), common DC progenitors (CDPs), and Pre-DCs (GSE60781) (*Schlitzer et al., 2015*). We used the FPKM table provided by the authors. After filtering, 242 single-cell samples and 5,489 genes are used in downstream analysis. The closest pair of groups, which consists of 59 MDPs and 89 CDPs was used for the unsupervised clustering comparison.

### Human induced pluripotent stem cell lines

scRNA-Seq data were extracted from induced pluripotent stem cell lines of three Yoruba individuals. On each individual three technical replicates were performed. A total of 5 bp

random sequence UMI were attached to the transcripts in order to avoid amplification bias. We used the molecule counts provided by the authors. After filtering, 864 single-cell samples and 9,750 genes are used in downstream analysis. The closest pair of groups, which consists of 288 cells from individual NA19098 and 288 cells from individual NA19239 was used for the unsupervised clustering comparison.

## Single-cell RNA-Seq analysis
### Read alignment
We downloaded the public datasets from NCBI The Gene Expression Omnibus (GEO) database (*Edgar, Domrachev & Lash, 2002*; *Barrett et al., 2013*), and retrieved the SRA files from The Sequence Read Archive (SRA) (*Leinonen et al., 2011*). We used the *fastq-dump* tool from SRA Toolkit to convert the SRA files into two pair-end FASTQ files. We applied *FastQC* for quality control and TopHat2 (*Kim et al., 2013*) for alignment to the reference genomes. The ready-to-use genome sequences and annotation files were downloaded from Illumina iGenomes page (http://support.illumina.com/sequencing/sequencing_software/igenome.html). For human build hg19 was used, and for mouse genome build mm10 was used (*Karolchik et al., 2014*).

### Read counting
We used *featureCounts* (*Liao, Smyth & Shi, 2014*) to map and count the aligned BAM files to the RefSeq transcriptomes from the pre-built packages on Illumina iGenome website above. We used the options to count fragments instead of reads; paired-end distance was checked by *featureCounts* when assigning fragments to meta-features or features. We only took into account of fragments that have both ends aligned successfully and discarded chimeric fragments. Fragments mapped to multiple locations were counted. The command is "featureCounts -pPBCM –primary -T 6 -a <gtf_ file>-o <output_ file><bam_ file>."

## Normalization of counts
We used reads per kilo base per million (FPKM) to represent the gene expression level, where the length of each gene was calculated by UCSC RefSeq annotation table, by concatenating all the exons. We normalized the data using DESeq2.

## Sample and gene filtering
Samples expressing housekeeping genes with geometric average FPKM lower than 4 were deemed abnormal and removed. Genes that were not expressed at all in over 70% of the cells were removed.

## Closest pair identification
The closest pair of groups among the samples is defined as the pair that has the smallest distance between the two groups' centroids, on the 2-dimensional correlation t-SNE plot. The correlation t-SNE is the t-SNE method (*Van der Maaten & Hinton, 2008*) performed on the correlation distance between two samples $x$ and $y$:

$$d_c(x,y) = 1 - cor(x,y).$$

## Non-negative matrix factorization (NMF)

We used the R-package implementation of NMF (*Gaujoux & Seoighe, 2010*) to perform NMF analysis. NMF is mathematically approximated by: $A \approx WH$, where $A$ ($n$ by $m$) is the matrix representing the scRNA-Seq profile in this report, $W$ is a slim weight matrix ($n$ by $k$, where $n \gg k$), $H$ is a wide matrix ($k$ by $m$, where $m \gg k$), and all three of them are non-negative (*Brunet et al., 2004*). The column vectors in $W$ are called *meta-genes*, which are higher-level abstraction of the original gene expression pattern. For gene $i$, the *loadings* are the $k$ values in $W$ at row $i$. We used the method "*brunet*" to solve the approximation of $A$, which employs the multiplicative iterative algorithm described by the following rules:

$$H_{au} \leftarrow H_{au} \frac{\sum_i \frac{W_{ia} V_{iu}}{(WH)_{iu}}}{\sum_k W_{ka}}$$

$$W_{ia} \leftarrow W_{ia} \frac{\sum_u \frac{H_{au} A_{iu}}{(WH)_{iu}}}{\sum_v H_{av}}.$$

The initialization of $H_{au}$ and $W_{ia}$ was generated as random seed matrices drawn from a uniform distribution within the same range as the entries in the matrix $A$. Since the starting matrices were randomized, we conducted an average of 30 simulations for each NMF run to obtain the consensus clustering results. We used Kullback–Leibler divergence (KL-divergence) as the distance function, as it has significantly better performance theorized in *Yang et al. (2011)*. The clustering results of all possible $k$'s (usually ranging from 2 to 5, as higher $k$ values require exponentially more time to run) were listed and $k$ was chosen when the best cophenetic correlation coefficient is achieved, as proposed in *Brunet et al. (2004)*.

### D-score

To rank the importance of genes using the two matrices factorized by NMF, we define D-score as follow: The D-score for gene $i$ is defined as:

$$D_i = \left| \lambda W_{i,1} - (1 - \lambda) W_{i,2} \right|.$$

The balancing factor $\lambda$ is determined by the rule that a gene uniformly expressed across all samples should have D-score 0. This score is a slight modification of the gene ranking method using discriminant NMF proposed earlier (*Jia et al., 2015*).

## Other packages used for detecting significant or important genes

*DE methods for bulk-level RNA-Seq*: we used two most popular bulk-level RNA-Seq methods: DESeq2 and edgeR, to compare on the results of DE genes.

*DE methods for scRNA-Seq*: three methods were investigated, with default settings of the packages: (1) Monocle: this is a versatile method (V 1.0.0) that performs differential expression analysis between cell types or states, moreover places cells in order according to their progression through processes such as cell differentiation (*Trapnell et al., 2014*); (2) SCDE: this package (V 1.2.1) implemented in R is based on Bayesian method, where the individual genes were modeled explicitly as a mixture of the dropout and amplification events by the Poisson model and negative binomial model (*Kharchenko, Silberstein & Scadden, 2014*); (3) MAST: this method (V 1.0.1) implemented in R was originally used to

detected DE genes in qPCR results of single cells. We selected the 500 genes with the lowest likelihood ratio test *p*-value using Hurdle Model provided by the package, as recommended by the authors (*McDavid et al., 2013*).

Additionally, we experimented if introducing t-SNE, a dimension reduction method that was recently successfully applied to scRNA-Seq, would improve the results of NMF. We used the C++ accelerated R-package Rtsne (V 0.10), based on the original C++ implementation (*Van der Maaten, 2013*).

## Module detection package

We used Functional Epigenetic Modules (FEM) R package (*Jiao, Widschwendter & Teschendorff, 2014*) for module detection. FEM utilizes an expansion algorithm based on the z-score of the expression level, by using a list of seed genes as the starting point. It selects the top modules based on *p*-values calculated by a Monte Carlo method.

We modified the source code of the FEM package and changed the process of the seed gene selection. Instead of selecting the seed genes based on the z-score of the expression level, we directly plugged in a list of genes as the seed genes, which were generated from each of the compared method for important gene detection.

## Measuring the performance of unsupervised clustering methods

Pair-wise Rand measure for clustering between the test and the reference is defined by

$$R = \frac{TP + TN}{TP + FP + FN + TN}$$

in which the four quantities *TP*, *FP*, *FN*, and *TN* are cardinals of the four sets of pairs. *T/F* means true/false based on the reference, and *P/N* means positive/negative results from the test. Specifically, a positive result (*P*) refers to a pair of samples clustered in the same group by the tested method; a true positive (*TP*) or true negative (*TN*) result represents the case where the agreements between the test and the reference clustering is reached (*Rand, 1971*).

## Modularity detection and pathway analysis

We used Functional Epigenetic Modules (FEM) package (*Jiao, Widschwendter & Teschendorff, 2014*) implemented in R for module detection. FEM utilizes *SpinGlass* algorithm (*Reichardt & Bornholdt, 2006*) based on the z-score of the expression level, by using a list of seed genes as the starting points. It selects the top modules based on *p*-values calculated from a Monte Carlo method. We modified the source code of the package to allow seed genes generated from other methods (NMF, DESeq2, edgeR, SCDE, MAST and Monocle) that detect significant or important genes. In each case, we used top 500 most important genes as the seeds for FEM. We next compared biological meanings of the resulting modules by Gene Ontology (GO) or Kyoto Encyclopedia of Genes and Genomes (KEGG) pathway enrichment analysis, implemented as DAVID Web Service in R (*Huang, Sherman & Lempicki, 2008*; *Huang, Sherman & Lempicki, 2009*).

## Data and code availability

The Glioblastoma, mouse lung distal epithelial, mouse bone marrow, and human induced pluripotent stem cell lines datasets are downloaded from Gene Expression

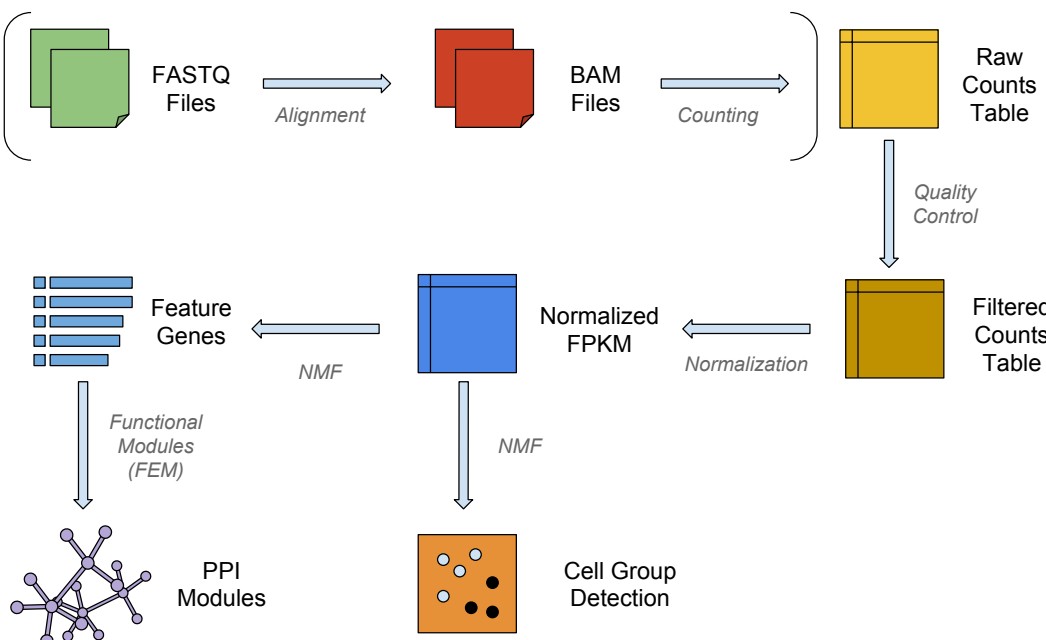

**Figure 1** **The workflow of NMFEM.** The input can be either FASTQ files or a raw counts table. If FASTQ files are used, they are aligned using TopHat and counted using FeatureCounts (steps shown in brackets). The input or calculated raw counts table are filtered by samples and genes, converted into FPKMs using gene lengths, and normalized by samples. We then run NMF method on them to detect groups of cells, and find the feature genes separating the detected groups. Finally, we feed the feature genes as seed genes in FEM, and generate PPI gene modules that contain highly differentially expressed genes.

Omnibus (GEO) repository with accession codes GSE57872, GSE52583, GSE60781, and GSE77288, respectively. The code used for the package NMFEM can be found at https://github.com/lanagarmire/NMFEM, and https://github.com/lanagarmire/NMFEM_extra.

## RESULTS

The workflow for a typical single-cell analysis using NMF is shown in Fig. 1. Briefly, the pipeline can take raw reads from FASTQ files, align and count them to the transcriptome; or it can use raw count data directly as the input matrix. The input data matrix is then subject to quality control and normalization steps. NMF then operates on the normalized matrix, clusters the samples into subpopulations (or subclasses) and enlists the feature genes that separate the subpopulations. In order to display the insightful biological modules and hotspots in the interactome, the feature genes are then used as seeds for a functional modularity detection algorithm FEM (*Jiao, Widschwendter & Teschendorff, 2014*).

### NMF accurately clusters scRNA-Seq data among similar populations

We first assess NMF's accuracy in unsupervised clustering compared to two other commonly used methods: K-means and hierarchical clustering (Hclust) algorithms. For generality, different distance metric and linkage variations of Hcust clustering were explored, including the combinations of Euclidean Distance & Complete Linkage (Euclidean + Complete), Euclidean Distance & Ward Linkage (Euclidean + Ward), and

Correlation Metric & Complete Linkage (Correlation + Complete). To draw unbiased conclusions, we tested these clustering methods on five datasets, including a mouse embryonic lung epithelial dataset (*Treutlein et al., 2014*), a mouse hematopoietic stem cell (HSC) and multipotent progenitor (MPP) cell dataset (Table S1), a human glioblastoma dataset (*Patel et al., 2014*), a mouse bone marrow dendritic cells dataset (*Schlitzer et al., 2015*), and lastly a human induced pluripotent stem cell (iPSC) dataset with unique molecular identifier (UMI) counts (*Tung et al., 2016*). A good clustering method should be able to separate groups that are most similar, therefore we selected the closet two groups in each dataset (Methods).

As shown in Fig. 2, NMF consistently achieves the highest performance among all methods compared, as measured by the Rand Index measure (Methods). Indeed, despite the high level of similarity between each pair of groups, NMF achieves an average Rand measure of 0.88 among the five datasets. In particular, it has an impressive Rand measure of 0.95 in the mouse embryonic lung epithelial dataset (Fig. 2A) and almost 1.00 in the UMI iPSC dataset (Fig. 2E). In contrast, K-means and the three variations of hierarchical clustering (Euclidean + Complete, Euclidean + Ward, and Correlation + Complete) all have significantly ($p$-value $< 0.05$) lower averaged Rand measures of 0.66, 0.52, 0.66, and 0.52, respectively (Fig. 2). Most of the wrongly clustered samples by K-means are located between the two clusters in the Euclidean space (Fig. S1A ). And hierarchical clustering methods can be heavily affected by individual outliers (Figs. S1B–S1D).

t-SNE is a commonly used dimension reduction method that exaggerates the differences between populations (*Van der Maaten & Hinton, 2008*), and was previously used in combination with other clustering methods on scRNA-Seq data (*Van der Maaten & Hinton, 2008*; *Bushati et al., 2011*; *Junker et al., 2014*). Therefore, we investigated the effect of t-SNE preprocessing on NMF and other compared methods next. Interestingly, NMF performs significantly worse after t-SNE preprocessing (Fig. 2). We speculate that the more significant decrease in accuracy by NMF is attributed to the fact that NMF is not a distance-based method, and the reduction of features after t-SNE interferes with NMF's ability to conduct component decomposition more drastically than other methods.

## NMF discovers biologically meaningful genes to separate subpopulations

One major advantage of NMF over many other unsupervised clustering methods such as K-means, is its ability to simultaneously identify genes that are characteristic to each subpopulations. We use the discriminative metric called D-score, based the previous rank-2 discriminant NMF implementation (*Jia et al., 2015*). Briefly, D-score is a non-negative number, and the more positive a gene's D-score is, the more unique this gene is expressed in a subpopulation (Methods). We ranked the top genes in NMF by D-score in the descending order, and sought prior literature reports to validate the utilities of NMF empirically.

For the top ranked NMF genes differentiating E14.5 and E16.5 (*Treutlein et al., 2014*), we found biological evidence to support their functions. For example, *Mal, T-Cell Differentiation Protein 2* (Mal2) ranked 2nd by NMF, was noted to be involved in the dynamic circuit of early lung development (*Ye, Liu & Wu, 2012*). Another gene

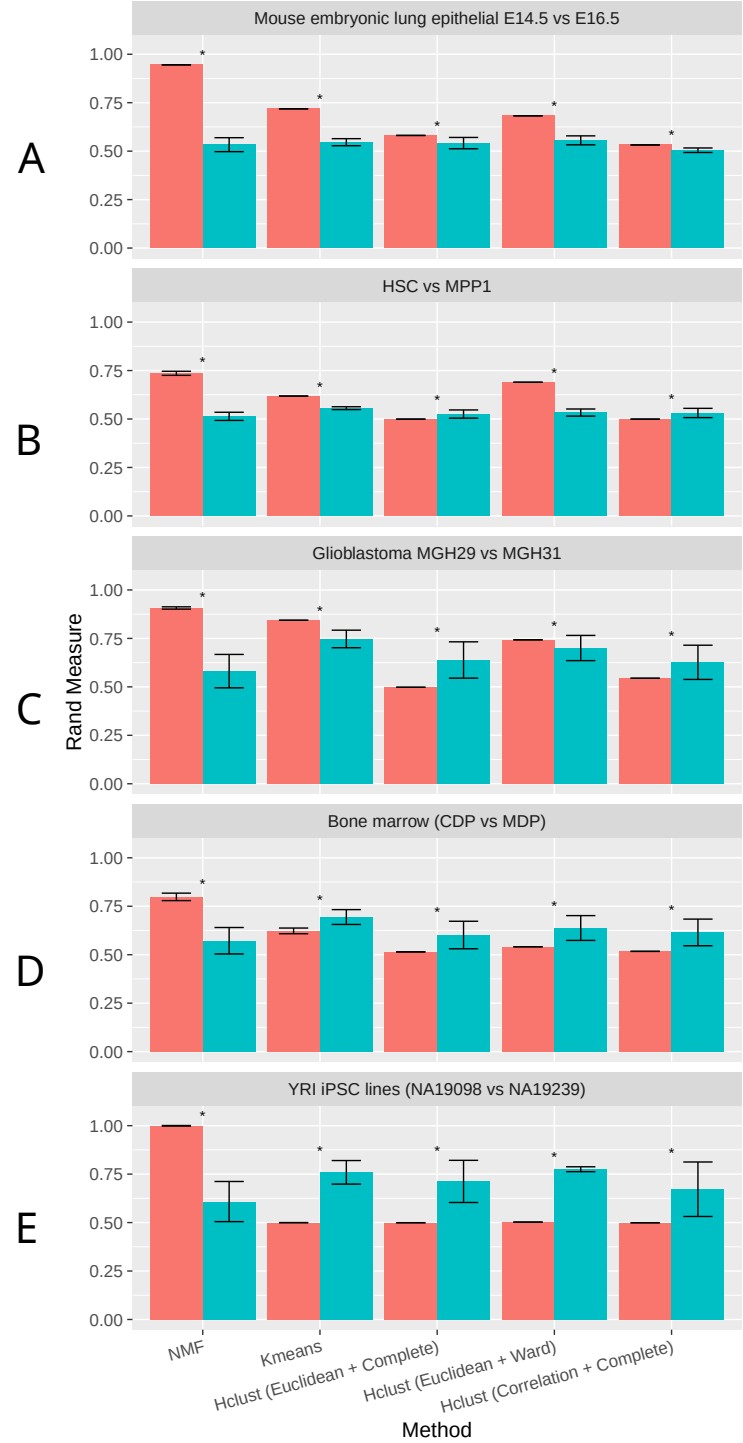

**Figure 2  Rand measures comparison of all methods on five datasets.** (A) Mouse embryonic lung epithelial E14.5 vs E16.5 (B) HSC vs. MPP1 (C) Glioblastoma MGH29 vs MGH31 (D) Bone marrow dendritic cells (CDP vs MDP) (E) human induced pluripotent stem cell (iPSC) lines with UMI counts. Rand measure ranges from 0 to 1, where a higher value indicates a greater clustering accuracy. The error bars show the standard deviation across 30 runs. Results significantly worse than NMF without tSNE by Welch $t$-test are marked by asterisks. For datasets with more than two groups of cells, the closest pair is selected.

*Surfactant Associated 2* (Sfta2) ranked 4th by NMF, was recently identified as an expression QTL (eQTL) gene during early lung development (*George et al., 2016*). Additionally, we examined top genes differentiating the subsets of AT1 vs. AT2 samples as reported originally (*Treutlein et al., 2014*). Impressively, the four genes Hopx, Ager, Egfl6, and Sftpc which were marker genes reported in the original report, have ranks of 1st, 12th, 23rd and 27th among the top genes according to NMF.

Similarly, in the HSC vs. MPP1 dataset, many of the top-ranked genes were previously noted as either differentially expressed between the two types of cells or characteristic to one of the two types. For example, *Cysteine Rich Protein 1* (Crip1), ranked 2nd by NMF D-score, is found to be expressed lower in Hdac3 knock-out cells undergoing HSC and MPP differentiation (*Summers et al., 2013*). *Regulator of G-Protein Signaling 2* (Rgs2), ranked 3rd by NMF, was noted to be differentially expressed in HSCs (*Phillips et al., 2000*; *Li & Akashi, 2003*; *Park et al., 2003*). *SKI-Like Proto-Oncogene* (Skil), ranked 8th by NMF, was shown to play important roles in hematopoietic development (*Pearson-White et al., 1995*).

For the MGH29 and MGH31 glioblastoma patient data (*Patel et al., 2014*), although we could not obtain definite clinical subtypes based on the original paper, we did observe that many top genes identified had previously been related to various types of cancers. For example, *Chromosome 8 Open Reading Frame 4* (C8orf4), ranked 9th by NMF, was identified as ''*Thyroid cancer 1* (TC1)'' (*Panebianco et al., 2015*; *Zhang et al., 2015*; *Zhu et al., 2015*; *Huang et al., 2016*). *Epidermal Growth Factor Receptor* (EGFR), a gene ranked 31st by NMF, has elevated expression from MGH29 to MGH31. EGFR is a well-known growth factor receptor noted for its critical role on cell division control. Moreover, three top genes EZR (ranked 12th), TUBB (ranked 27th), and RDX (ranked 36th), were shown to be involved in organization and regulation of morphological characteristics of breast cancer cells including cell shape and membrane to membrane docking (*Kopp et al., 2016*).

## Comparison of important genes from NMF and other methods

To further understand the patterns recognized by NMF, we compared the top ranked genes identified by NMF with the those by other recent scRNA-Seq analysis methods, including Monocle (*Trapnell et al., 2014*), MAST (*McDavid et al., 2013*) as well as SCDE (*Kharchenko, Silberstein & Scadden, 2014*) (*Treutlein et al., 2014*). In addition, we included DESeq2 and edgeR, two differential expression (DE) methods commonly used in bulk RNA-Seq analysis.

We compared the different methods using MA plots to analyze the mouse embryonic lung epithelial cells dataset (E14.5 and E16.5, Fig. 3). For fair-comparison, we highlighted the top 500 genes identified by each method in blue. Since the global gene expression levels are reduced from E14.5 to E16.5, all methods detect mainly down-regulated genes. The specific genes identified by these methods however vary greatly (Fig. 3, Figs. S2 and S3), but NMF is most different from other methods, based on its unique mathematical formulation. Unlike other methods that are prone to lowly expressed genes, NMF tends to select genes that have intermediate expression levels (Figs. S3). Indeed, 90% of selected genes have log FPKM between 2.32 and 4.66. Other methods, on the other hand, select more genes with lower expression levels. Specifically, SCDE, MAST, DESeq2, edgeR and Monocle have 56%,

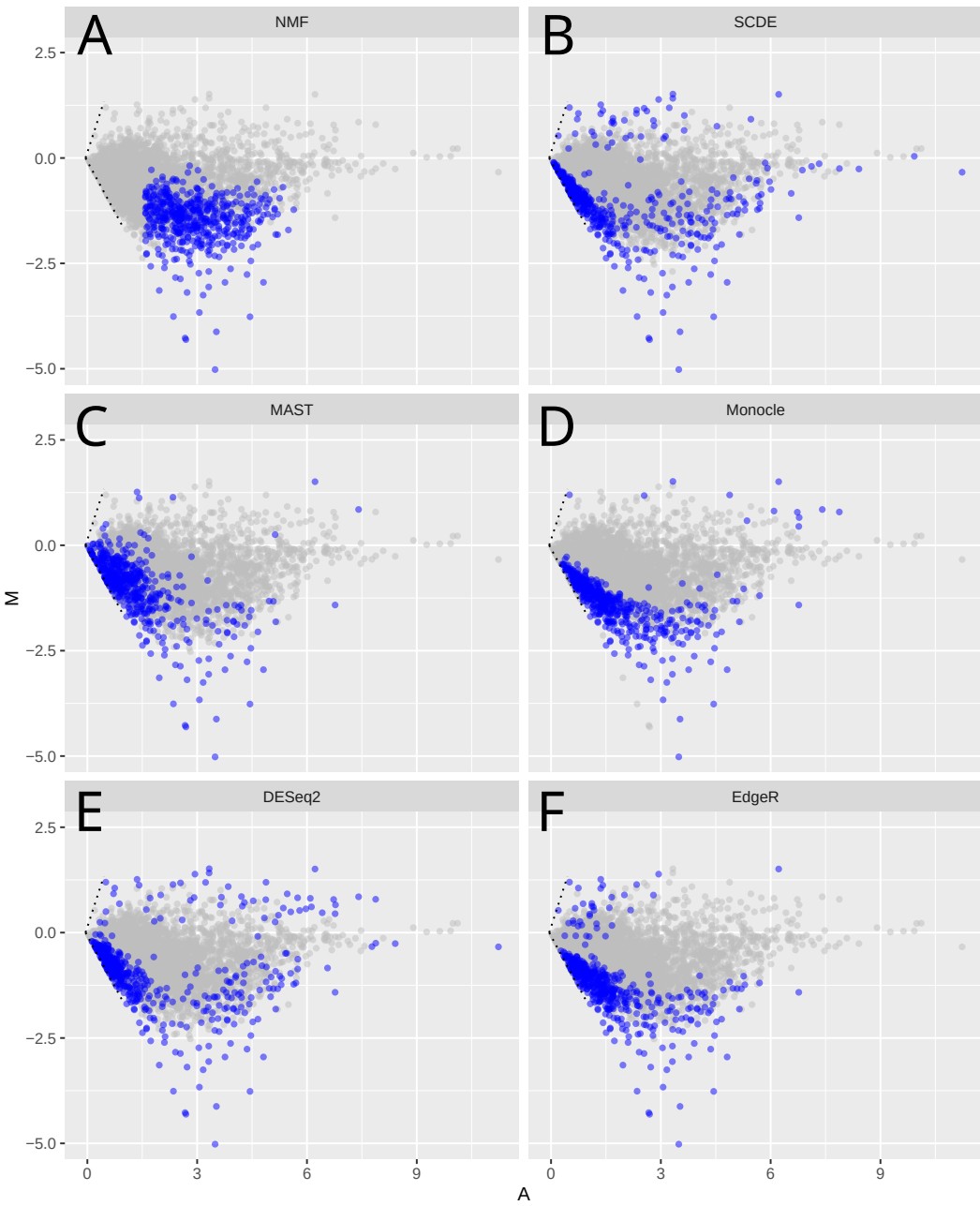

**Figure 3** **MA-plots of significant or important genes identified by different methods.** Shown are scRNA-Seq data in the mouse lung distal epithelial cell E14.5 vs. E16.5 samples. The blue color highlights the genes selected as "the most significant" by the corresponding methods. $X$-axis ($A$-value) is the mean of the gene expression, and $y$-axis ($M$-value) is the difference of the gene expression between E16.5 and E14.5 stages.

45%, 31%, 27% and 18% genes below FPKM $= 1$, respectively. The same pattern has been observed in the HSC vs. MPP, glioblastoma, and bone marrow dataset (Figs. S4A–S4C). Lowly expressed genes usually have relatively higher levels of noise (*Brennecke et al., 2013*), evident by comparing the CV2 levels between genes selected by different methods (Fig. S5), and their apparent differential expression may be less reliable.

We further tested whether such intermediately expressed genes identified by NMF are robust and unlikely a result of random sampling. We performed additional runs with each run deliberately dropping one sample from the dataset. The results show that the top 500 genes have over 95% probability to re-appear in this top 500 list, compared to less than 2% probability for any other non-top genes (Fig. S6). The reason that NMF tends to avoid lowly expressed genes is that D-score considers absolute expression level of a gene, rather than the relative expression level (see 'Methods'). The lower its expression level, the less likely a gene is selected as a feature gene by NMF. On the other hand, very highly expressed genes are more likely to be expressed across all cells. They do not generate large differences in their meta-gene loadings, and will not achieve high D-score as well.

### Modularity detection with NMF based framework

We next asked if the important genes detected by NMF convey unique and meaningful biological functions. Previously, a community detection method based on *SpinGlass* algorithm named FEM has been reported to detect functional modules on protein-protein interaction (PPI) networks (*Jiao, Widschwendter & Teschendorff, 2014*). Inspired by this approach, we used the 500 top genes selected by NMF as seed genes, and ran vertex-initialized *SpinGlass* models on the PPI network. Modified from FEM, here each vertex (gene) is weighted by $t$-statistics measuring the expression level (rather than DNA methylation) changes. In order to assess the robustness of the method, we performed random permutations on the data and calculated the $p$-value for each module. The top four modules for mouse embryonic lung epithelial cells dataset are shown in Fig. 4. We conducted the same procedure on HSC vs. MPP1, glioblastoma, and bone marrow datasets (Figs. S4D–S4F).

Consistent with the down-regulating trend of all genes from E14.5 to E16.5 (Fig. 3), these top four modules exhibit strong down-regulation as well. Five hub genes are identified: *Rad21*, *Atf3*, *Polr2j*, *Tecr* and *Rpl37*, among which *Tecr* and *Rpl37* are in the same module. These modules are involved in chromosome structural maintenance (seed *Rad21*), transcription activation (seed *Atf3*), RNA polymerase (seed *Polr2j*), and ribosome component and various enzyme coding genes (*Tecr* and *Rpl37*). The fact that the top modules are all related to transcription and translation is interesting, as it explains the observed global gene expression down-regulation observed earlier (Fig. 3).

## DISCUSSION & CONCLUSIONS

Due to the high level of noise in scRNA-Seq data (*Brennecke et al., 2013*) and its unique challenges to elucidate the subtle relationships of a seemingly similar single cells in a population, the traditional DE approaches that rely on known labels and strong expression differences of single genes may be limited. Meanwhile, NMF with the ability to capture

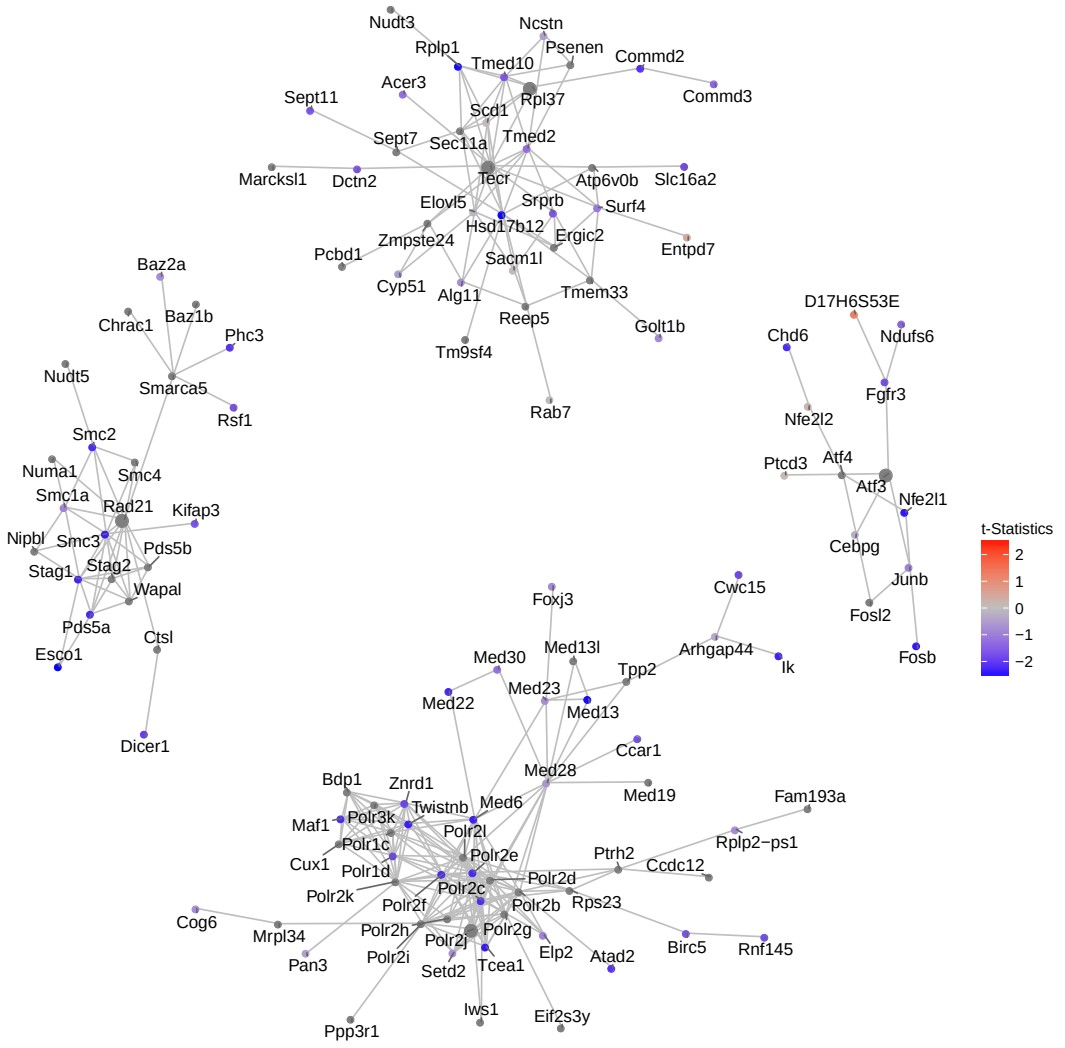

**Figure 4 Network of top 5 modules using the seed genes generated by NMF.** Shown are module detection results in the FEM package, using the top 500 most important genes detected by NMF in Fig. 3. scRNA-Seq data in the mouse lung distal epithelial cell E14.5 vs. E16.5 samples are compared, where the red and blue colors indicate up- and down-regulation of genes in E16.5 relative to E14.5, respectively. The top five modules are selected by the $p$-values calculated from the internal Monte Carlo method in the FEM package.

the general expression pattern changes may be very suitable for scRNA-Seq subpopulation identification. Previous applications of NMF to fields such as face reorganization (*Rajapakse, Tan & Rajapakse, 2004*), image compression (*Yuan & Oja, 2005*; *Monga & Mıhçak, 2007*) and sound decomposition (*Smaragdis, 2004*), have been proven successful. Here we propose to utilize NMF as a suitable method for scRNA-Seq analysis.

Specifically, we have demonstrated that in multiple datasets and multiple scRNA-Seq protocols, NMF performed significantly better than other popular clustering methods including K-means and hierarchical clustering. Moreover, since NMF is an unsupervised method, it is not affected by preconceived cell type labeling. Its higher accuracy and

additional ability to extract feature genes may allow it to serve as the direct substitute to other commonly used unsupervised methods. It should be noted however, that this benefit of NMF diminishes when the classes are drastically different. Also, the fact that it has the tendency to select more highly expressed and correlated genes implies that it might miss biologically important genes (such as transcription factors) that have low expression levels (*Tian et al., 2011*).

Another caveat of our secondary data analysis on public data set, is that there are batch effects that confound the patterns recognized by NMF (or any other subpopulation detection methods). In these experiments, groups of samples are sequenced in different runs, thus technical variation may confound the biological variation (*Hicks, Teng & Irizarry, 2015*). We calculated the confoundedness of the three datasets (mouse embryonic lung epithelial, glioblastoma and bone marrow datasets) where experimental information is available, and found that confounding accounts 92.8%, 98.9% and 100%, respectively. However, since NMF is able to identify some previously validated biological results, it is reasonable to believe that NMF still has the power to detect biological factors. It is nonetheless paramount to design experiments well so that such confounding effects can be minimized in the future scRNA-Seq studies.

Moreover, the best combinatory approaches for scRNA-Seq subpopulation identification will continue to be an interesting area of research. For example, a few normalization methods have been proposed for RNA-Seq experiments lately, including RUV (*Risso et al., 2014*) and GRM (*Ding et al., 2015*). These normalization methods may have additional impact on the results of clustering. More rigorous source of variation (SOV) analysis to identify the best combinations of analysis steps will be an interesting follow-up project.

In summary, we have demonstrated that NMF is a method capable of accomplishing various tasks in scRNA-Seq data analysis, from reclassifying populations of single cells to revealing meaningful genes and modules of biological significance. We expect the new workflow proposed here will have valuable applications in the field of scRNA-Seq bioinformatics analysis.

## ACKNOWLEDGEMENTS

We would like to thank our colleagues Kumardeep Chaudhary and Olivier Poirion for reading the manuscript and providing valuable suggestions.

### Funding

This research was supported by grants K01ES025434 awarded by NIEHS through funds provided by the trans-NIH Big Data to Knowledge (BD2K) initiative (http://www.bd2k.nih.gov), P20 COBRE GM103457 awarded by NIH/NIGMS, NICHD R01 HD084633 and NLM R01LM012373 and Medical Research Grant 14ADVC-64566 from Hawaii Community Foundation to LX Garmire. The funders had no role in study design, data collection and analysis, decision to publish, or preparation of the manuscript.

## Grant Disclosures

The following grant information was disclosed by the authors:
NIEHS: K01ES025434.
NIH/NIGMS: P20 COBRE GM103457.
Hawaii Community Foundation: NICHD R01 HD084633, NLM R01LM012373, 14ADVC-64566.

## Competing Interests

The authors declare there are no competing interests.

## Author Contributions

- Xun Zhu performed the experiments, analyzed the data, wrote the paper, prepared figures and/or tables, reviewed drafts of the paper.
- Travers Ching analyzed the data, reviewed drafts of the paper.
- Xinghua Pan and Sherman M. Weissman contributed reagents/materials/analysis tools, reviewed drafts of the paper.
- Lana Garmire conceived and designed the experiments, analyzed the data, wrote the paper, reviewed drafts of the paper.

## Data Availability

GitHub: https://github.com/lanagarmire/NMFEM.

## Supplemental Information

Supplemental information for this article can be found online at http://dx.doi.org/10.7717/peerj.2888#supplemental-information.

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
