# Peer review of "Detecting heterogeneity in single-cell RNA-Seq data by non-negative matrix factorization"

_PeerJ, doi:10.7717/peerj.2888_

## Round 0.1 · original submission · Major Revisions

While the reviewers found the topic interesting, they have raised many concerns about the competing methods (Hclust), benchmarking data sets, and the presentation of the method and results. Please revise according to the reviewer comments.

·

Basic reporting

In this report, Zhu et al. apply non-negative matrix factorization (NMF) to analyze multiple single cell RNA-Seq datasets to cluster single cells in an unsupervised manner, stratify subpopulations, and detect modules associated with differences among subpopulations.

In the introduction, the authors present a good overview of previous discoveries from single cell RNA-seq data and aptly suggests a need for more bioinformatics tools. However, the authors are missing a number of papers in the scRNA-seq analysis domain of relevance to their method, including
- http://www.nature.com/nbt/journal/v33/n2/full/nbt.3102.html (scLVM)
- http://science.sciencemag.org/content/347/6226/1138 (BackSPIN)
- http://www.nature.com/nmeth/journal/v13/n3/abs/nmeth.3734.html (PAGODA)
- http://www.satijalab.org/seurat.html (SEURAT)
All these methods have been proposed for detecting subpopulations from single cell RNA-seq data. Furthermore, both BackSPIN and PAGODA identify genes, gene sets, and pathways (modules) associated with differences among subpopulations. While I do not believe a direct comparison of all these methods to NMF is within the scope of this paper, they are worth noting in the introduction and statements such as "However, it is not clear if all these new methods are suitable for detecting subpopulations in single cells. Moreover, none of the packages mentioned above offers functionalities for modularity identification." should be revised accordingly.

The authors combine NMF with a modified, seed based module detection tool and offer the resulting code as an R package called NMFEM. However, I have checked out the NMFEM code on Github. NMFEM in its current state does not constitute an R package and is rather a collection of R scripts. I strongly recommend that the authors follow Hadley's R Packages guide (http://r-pkgs.had.co.nz/) to turn these scripts into a real R package to be more useful for the community.

Experimental design

The authors compare the accuracies of NMF in unsupervised clustering to K-means and hierarchical clustering (Hclust) algorithms, benchmarking on two published datasets.

For hierarchical clustering, it is not clear what linkage criterion (complete, ward) and distance metrics (euclidean, correlation) were used. The author's should clarify. A number of linkage criterion and distance metric combinations should be assessed if the authors propose that Hclust is in general worse than NMF as suggested in the current manuscript. Correlation with Ward clustering and Euclidean distance with Complete clustering are just a few common combinations worth comparing.

The authors suggest that NMF has higher accuracy over K-means and Hclust based on the Rand-index measure for separating HSC and MPP cell types. However, it is unclear whether the same trends will hold for different cell types or more complex or subtle subpopulation differences. The authors analyze another dendritic cell differentiation data, with similar but less dramatic results. Because it is not clear whether the advantage they saw with these 2 datasets would generalize, the authors should analyze additional public datasets where cell annotations are available and assess performances.

The authors demonstrated that NMF performs well relative to other popular clustering methods such as K-means and Hclust. However, the authors do not compare NMF to more dedicated single cell subpopulation discovery methods as noted previously. Again, I do not believe it is within the scope of this paper to perform a direct comparison with all these other methods. However, I would like to see a discussion on the limitations of NMF, its advantages, and recommendations to users as to when to use NMF.

Validity of the findings

In assessing the accuracy of NMF compared to other algorithms, the authors benchmark on datasets comprised of multiple known cell types. However, it is not well described in the current manuscript whether these cell types were sequenced in the same of different batches. If these cells are from different batches, how do you know if NMF is truly better at distinguishing based on cell type and not just better at picking up batch effects? Please see http://biorxiv.org/content/early/2015/12/27/025528 for the impact of batch on scRNA-seq.

Again, this concern will be addressed once the authors analyze at least a few additional public datasets as part of them benchmarking as recommended previously. In particular, I recommend including a UMI dataset as part of the benchmarking since these datasets have distinctly different degrees of technical noise so it will be interested to see if the authors' claims generalize to single-cell RNA-Seq with UMI.

The authors apply NMF on GBM data from Patel et al. to find subpopulations within the same tumor. In the original publication, Patel et al. also suggest that there are two distinct genetic subpopulations in MGH31 marked by different CNVs. How do these genetic subpopulations correspond to the transcriptional subpopulations identified by the authors here? Is there overlap?

Also, in the analysis on GBM data, in Supp Fig. 6B, the PCA plot shows a distinct outlier cell driving PC2. What happens if this cell is removed?

The authors show that SCDE, MAST and Monocle have more similar results while DESeq2 and EdgeR have more similar results, speculating that this is due to the methods being developed for scRNA-seq and bulk RNA-seq data respectively. However, it appears that NMF has more similar results to DESeq2 and EdgeR as seen in Supp Fig 5, suggesting that it may be more appropriate for bulk cell RNA-Seq data and not for scRNA-seq. How do the authors resolve this result?

The authors note that the number of important genes identified by SCDE, MAST, NMF, etc vary greatly. How does this base number of important genes identified influence the degree of overlap and agreement between methods? For example, hypothetically, if NMF is very conservative and only identified 10 important genes, while DESeq2 is very liberal and identified 1000 significantly differentially expressed genes, the maximum overlap between these two methods is 10. Alternatively, if the authors select the same number of top genes from each method, say 500, the comparison may not be valid since by NMF 490 of those genes were not significant anyway.

Additional comments

Some specific comments on language:
- Line 137 - "bulky RNA-Seq data" should be "bulk RNA-seq data"
- Line 145 - With regards to "NMF selects genes that are sufficiently expressed in many samples, with a strong preference to select genes around a specific expression level (FPKM 2.740) and but not genes expressed too lowly or too highly" can you provide a more quantitative description of the findings? ex. 95% of genes selected by NMF fall within FPKM range X and Y.

Some specific comments on figures:
- Fig 4. Many of the gene names are not legible due to overlaps. I recommend either manually moving the nodes to be more spread out, or using a package like ggrepel to be able to automatically repels nodes away from each other for a more legible figure. The color choice is also difficult to read for red-green color blind individuals.

Reviewer 2 ·

Basic reporting

pass

Experimental design

No Comments

Validity of the findings

No Comments

Additional comments

The work is trying to development a new method for single cell RNA-seq analysis based on NMF. It is an interesting topic. However, the manuscript is hard to follow. First, in the introduction part, the authors reviewed both the methods for detecting differential expressed genes and detecting for sub-populations, it is confusing to the audience about the function of the new package. Second, Figure 2 A and B is hard to read, I think table is enough. Also, why the coordinates for figure 2A and 2B are different ? The label for figure 2C is wrong. Third, Why the authors demonstrate the different functions of NMF package with different data ? In the Results part, Part 1 is the performance of cluster on the HSC data; Part 2 is the performance of gene detection on the lung development data. It is confusing, please show the performance of all functions on all the samples, otherwise, it looks like that the author only pick the good results. Fourth, the normalization methods are also important in cluster. Recently reports, RUVseq(Risso et.al. Nature biotechnology, 2014) and GRM (Ding et.al Bioinformatics, 2015) have showed that the cluster can be improved given properly normalization. So please discuss the performance of NMF on different normalization methods.
Please address the above issues for publication purpose.

·

Basic reporting

Thank you to the authors for a nice overview of their novel approach to scRNA-seq analysis. I have a few comments regarding the quality of figures and their labeling/descriptions. Many of the figures are not labeled in a way that is clear and easy to understand/intrepret. E.g. Fig 2 legends contain "Correct: TRUE vs FALSE" or "after t-SNE: TRUE vs FALSE". I feel that a much more clear presentation would be to change the legend label to something like "Correctly assigned vs Incorrectly assigned" and "Without t-SNE correction vs With t-SNE correction". The titles of each figure can also be made more descriptive, rather than just, e.g. "MA-plot". Ideally, figure titles should say what they show: "MA-plot showing significant genes uniquely identified by each method". I strongly encourage the authors to be more thorough and thoughtful about figure presentation, since many readers may only read the intro to the article, and review the figures without diving into the main text, if they are short on time. There are also a few spelling errors in the manuscript and figures (Fig 5A; line 177).

Experimental design

One major flaw in the experimental design is that the authors did not use the NMF method on more distinct subpopulations, and validate that the important genes found using NMF, has previously been biologically validated. e.g. in the lung dataset, can NMF correctly find the genes that are important between say, the AT1 and AT2 lineages that were analyzed in the original paper? It was found in the Treutlein paper that genes such as Hopx, Ager, Egfl6, and Sftpc (some of which were validated biologically in the past; some were validated in the paper using in-situ imaging) are important genes that DEFINE the two alveolar cell types AT1 and AT2. One of the key results of the Treutlein paper is using scRNA-seq to identify additional novel biomarkers to distinguish the two populations. Does NMF pick up the same important biologically-validated genes between these two populations? Although it is interesting that this new analysis can find important genes between two closely related populations, it would give the reader much more confidence in the authors' claims if the authors can first demonstrate that their method reproduces biologically-validated results in a robust manner between two distinct subpopulations. Similar comments apply for the analyses on glioblastoma and cells of the mouse hematopoietic lineage. A more minor comment: in the glioblastoma case, do the authors have clinical data to support their findings related to microbial/viral infection?

Validity of the findings

The data as presented look fine. One comment on Fig 2C - can authors provide p-values for the before/after t-SNE Rand scores? Text claims improvement, but I cannot tell from the figure whether the improvement is significant.

Additional comments

Although this is not something I would require in a revision, I am also curious as to whether the authors have thoughts on Fig 4 - there are overwhelmingly more down-regulated genes than up-regulated ones from E14.5 to E16.5. Why is that? Is this trend also reflected in differential gene expression analysis? Have the authors explored whether there could be a systematic preference of the NMF method to pick up down-regulation as opposed to up-regulation?

---

## Round 0.2 · Minor Revisions

The reviewers all found significant improvement on the previous version. However, they still have some concerns about the interpretation of "lower-expression level genes" and the overstatement of the advantages of the method. Please address these concerns in the discussion.

·

Basic reporting

The authors have made substantial improvements and have thoroughly addressed my concerns from the initial review.

Experimental design

I have tested out the package and it installs easily from Github. I appreciate the example and would recommend the authors provide a usage vignette, and submit to CRAN or Bioconductor to improve usability.

I appreciate the discussion on batch confounding. However, the lack of a dataset with less than 90% confounding greatly diminishes some of the findings. I agree, that while "NMF is able to identify some previously validated biological results, it is reasonable to believe that NMF still has the power to detect biological factors" but all findings would be enhanced by benchmarking on a dataset from the same batch. I would encourage the authors to look for one more dataset from the same batch where annotations are available if they wish to enhance the impact of their paper.

Validity of the findings

One interesting difference between NMF and all other tested methods is its tendency to select genes that higher expression levels. While I agree with the authors that "lowly expressed genes usually have much higher relative levels of noise", their differential expression may still be of biological interest. As a more concrete example, transcription factors are often very lowly expressed but their differential expression across conditions is of great biological interest. Would NMF miss transcriptional factors since they are lowly expressed? The limitations due to NMFs tendency to choose genes with higher expression levels should be discussed.

I disagree with the conclusion: "The biggest advantage of NMF is that it can discriminate the subpopulations even though the differences among them are subtle." There are numerous other methods noted in the introduction that can also do the same, albeit requiring longer run-time are more computationally intensive. The improved performance over K-means and hierarchical clustering along with the ability to extract out feature genes are sufficient as notable advantages.

Additional comments

There appear to be 2 different versions of Figure 2s being referenced in the text and in the legends:

v1. Before the figure: "Figure 2(on next page)
Comparisons among clustering methods on the HSC vs. MPP1 scRNA-Seq data
(A) The PCA scatter-plots of the samples, based on their log normalized expression level. Colors indicate the most favorable labeling that can be assigned to the clustering result generated by each method. The correctly and incorrectly labeled samples are marked by dot (•) and cross (x), respectively. Confusion matrices of the methods in comparison are inserted on the top-right corner of each sub-panel. The closer the matrix is to a diagonal matrix, the more accurate the method is. (B) The scatter-plots of the samples for K-means and hierarchical clustering, after t-SNE based dimension reduction. (C) Rand measures of the methods in comparison, before and after t-SNE. Rand measure ranges from 0 to 1, where a higher value indicates a greater clustering accuracy."

v2. From the Figure Legends section: "Fig. 2: Rand measures comparison of all methods on five datasets. (A) Mouse embryonic lung epithelial E14.5 vs E16.5 (B) HSC vs. MPP1 (C) Glioblastoma MGH29 vs MGH31 (D) Bone marrow dendritic cells (CDP vs MDP) (E) human induced pluripotent stem cell (iPSC) lines with UMI counts. Rand measure ranges from 0 to 1, where a higher value indicates a greater clustering accuracy. The error bars show the standard deviation across 30 runs. Results significantly worse than NMF without tSNE by Welch t-test are marked by asterisks. For datasets with more than two groups of cells, the closest pair is selected."

The "NMF accurately clusters scRNA-Seq data among similar populations" section appears to reference v2 of Figure 2.

I'm not sure why Figure 2 is being referenced in the "NMF discovers biologically meaningful genes to separate subpopulations" section, other than to refer to the dataset. Please cite the dataset's paper or refer to methods instead of referencing a figure if the intent is to refer to the dataset, though I'm guessing this confusion may be caused by the Figure 2 legend and image discrepancy.

Figure 2 v2 currently provides a good summary of the performances of various methods. However, a supplementary figure showing the actual hierarchical clustering for one of the datasets analyzed with incorrect branches marked, or K-means with incorrect cell grouping highlighted would be more illustrative.

Reviewer 2 ·

Basic reporting

No Comments

Experimental design

No Comments

Validity of the findings

No Comments

Additional comments

All the issues have been addressed properly. I think the paper is in a good shape for publication purpose

·

Basic reporting

No comments

Experimental design

No comments.

Validity of the findings

No comment.

Additional comments

General comments for the author:

Thank you to the authors for addressing my questions and concerns from the last round of review. The manuscript reads more smoothly now, and presents a more cohesive and convincing case for NMF as a useful tool in single cell RNA-seq analysis compared to before. I have just a few minor points remaining:

1) Line 274 – “We speculate that this decline in accuracy is due to at least two reasons. First, unlike K-means or Hclust, NMF is not a distance-based method, and second, the reduction of features after t-SNE hinders NMF’s ability to conduct component decomposition.”

I’m not sure that I really agree with the authors speculations here. If it is indeed true that the accuracy of the method falls with t-SNE correction because of NMF being not distance-based and due to t-SNE reducing the number of features, then we should not expect to see the accuracy fall for other methods that ARE distance-based and that DO benefit from a reduction in number of features. However, based on Fig 2 data, this is not the case. For k-means, which is distance-based, we also see reduction in accuracy after using t-SNE correction in 3 out of 5 sample sets tested. The same is true of Hclust using Euclidean+Ward distance metric. The speculated reasons stated by the author initially appear to be quite plausible from intuition; but examination of the data implies that there may be something else going on. I would suggest that the authors be cautious in this speculation.



2) Minor typo: line 310 – “…critical role on cell division control. Moreover, three genes top genes EZR (ranked 12th), TUBB…”



3) Line 326 – “Unlike other methods that are prone to lowly expressed genes, NMF tends to select genes that have intermediate expression levels (Fig. S2). Indeed, 90% of selected genes have log FPKM between 2.32 and 4.66. Other methods, on the other hand, select more genes with lower expression levels. Specifically, SCDE, MAST, DESeq2, edgeR and Monocle have 56%, 45%, 31%, 27% and 18% genes below FPKM=1, respectively. Since lowly expressed genes usually have much higher relative levels of noise (Brennecke et al., 2013), their apparent differential expression could be less reliable and less likely to be biologically meaningful.”

and

Line 338 – “…non-top genes (Fig. S4). The reason that NMF tends to avoid lowly expressed genes is that D-score considers absolute expression level of a gene, rather than the relative expression level (see Methods).”

I find this observation extremely interesting in the context of discussion on single cell RNA-seq data noise levels. Based on the work of several other groups (some which the authors have cited), including Marioni group, van Oudenaarden group, Sandberg group, and Kharchenko group, indeed the level of noise in the data, whether technical or biological, can affect the genes that are selected as significant in distinguishing different subpopulations of cells. I find the authors speculated explanation for why NMF may prefer to select intermediate expression-level genes while other methods prefer to select low expression-level genes, to be plausible. But, it would be nice to directly look at this. One method would be, instead of looking at just the mean expression level of the genes selected by each method, to also compute the noise level of those genes, and see whether different methods are selecting based on noise level. A simple way to do this is to just calculate the CV or MAD for each gene that is selected, and plot distributions of the CV values, and compare the distributions for each method. This would directly evaluate the sensitivity of each method to the level of noise.
Furthermore, I think that it may be dangerous to claim that NMF is a better method simply because it avoids picking low-expression level genes. As others have found in several biological systems, expression of important cell-type determining genes could be quite low but important/consistent. LGR5, OLFM4, and TERT expression in intestinal and colon stem cells is a good example. It may be helpful to include a caveat on the benefit of using NMF in this regard.


4) The authors do a good job of demonstrating the advantages of using NMF for single-cell RNA-seq analysis. But what about the caveats, disadvatanges, or situations where NMF may not be appropriate? In particular, since many people who do scRNA-seq analysis may not be bioinformatics or statistics experts, it may be helpful for the authors to summarize and discuss what kinds of application they feel the methods is suitable or not suitable for. All computational methods will have situations in which it is less appropriate than others. It would be nice to see a discussion of this kind in the manuscript as well, and I do not think it would detract from the usefulness of the tool presented by the authors.

---

## Round 0.3 · accepted · Accept

This manuscript is now in good shape and ready for publication.